# Anthropometric Profile and Physical Fitness Performance Comparison by Game Position in the Chile Women's Senior National Football Team

**Rodrigo Villaseca-Vicuña** [1,2], **Edgardo Molina-Sotomayor** [3], **Santiago Zabaloy** [2,4]
**and Jose Antonio Gonzalez-Jurado** [2,*]

1   Federación de Fútbol de Chile, Santiago 7930013, Chile; rvillaseca@anfpchile.cl or rvillasecav@gmail.com
2   Physical Performance and Sports Research Centre, University of Pablo de Olavide, 41013 Sevilla, Spain; santiagozabaloy@hotmail.com
3   Facultad de Artes y Educación Física, Departamento de Educación Física, Universidad Metropolitana de Ciencias de la Educación, Santiago 8320000, Chile; edgardo.molina@umce.cl
4   Facultad de Actividad Física y Deportes, Universidad de las Flores, Buenos Aires 1406, Argentina
*   Correspondence: jagonjur@upo.es; Tel.: +34-954-977-586

**Abstract:** The aim of this study was to explore the anthropometric profile and evaluate physical fitness variables of the members of the Chile women's national football team based on their playing positions. Fifty football players participated in this study, which was carried out during the period of training for the France 2019 Women's World Cup and the Japan 2020 Olympic Games. Body composition and physical condition (muscular strength, sprint, agility, and aerobic fitness) were assessed. The goalkeepers showed greater weight ($p < 0.001$), height ($p = 0.002$), and %Fat ($p = 0.010$) compared to the rest of the playing positions. There were also differences between positions in relative strength (RS) ($p = 0.001$), running speed at 10 and 30 m (T10 and T30, respectively), agility (AGI) ($p < 0.001$), and yo-yo test (MYYR1) ($p < 0.001$). RS, T10, T30, and countermovement jump (CMJ) were significantly correlated ($p \leq 0.05$) with anthropometric variables ($p \leq 0.05$). MYYR1 was also significantly correlated with anthropometric variables ($p \leq 0.05$). In conclusion, goalkeepers show greater weight and height, as well as worse results in MYYR1, AGI, T10, T30, and RS compared to the rest of the players. Forwards present better performance in running speed and agility. Better performance in physical condition is associated with better body composition values (greater muscle mass and lower fat mass). Greater relative strength indicates greater performance in explosive actions. Therefore, to meet the demands of high competition, it is important to establish ideal profiles in anthropometry and physical condition variables based on the playing position.

**Keywords:** physical performance; soccer; anthropometric profile; women's football

## 1. Introduction

From the physiological perspective, European football (soccer in USA) is understood as an intermittent activity that combines aerobic and anaerobic efforts at different intensity levels with irregular pauses [1,2]. Moreover, it is a team sport with unique performance characteristics and demands [1–3]. An official football match has a duration of 90 min, which can be extended to up to 120 min in some cases [4]. The effective playing time varies; e.g., in the France 2019 Women's World Cup, an average of 54 min per match was recorded [5]. During such time, different skills, such as sprinting, jumping, and changing direction, are combined with technical-tactical elements, such as dribbling, passing, and shooting, among others.

It is important to highlight that physical and technical demands are determined by the different positions or roles of each player in the team. In this sense, there are different classifications or types of playing positions in a football team. The most described positions are usually goalkeepers, defenders, midfielders, and forwards [3,6]. Thus, for example,

each player covers different distances according to her position [7]. In the analysis of the France 2019 World Cup, the average total distance covered was 5362 m for goalkeepers, 10,369 m for defenders, 11,210 m for midfielders, and 10,979 m for forwards. Regarding the intensity of the distances covered, for the medium–high range (19–23 km/h) the distances covered according to the playing position was 253 m for defenders, 313 m for midfielders, and 360 m for forwards [5]. In this sense, it has been reported that the amount of physical performance, effort intensity, and movement patterns are different depending on the playing position [8].

Anthropometry is the measurement methodology of the human body. Its objective is to quantify the morphological features and to provide an objective image of the growth status of the person. Morphological characteristics appear to be very important for selection in most sports disciplines, including European football. For sports disciplines such as European football, the morphological structure that affects sports performance is already known [9]. There are differences in body composition between periods of the season (higher values of adipose tissue in the transition period vs. the competition period) [10] as well as differences in anthropometric values between positions of game [11]. A study conducted with young male players reported that goalkeepers have statistically significantly higher body height and humerus width compared to players who play in other team positions and more weight compared to players who play in the middle and forward positions. Defense players have statistically significantly higher body weight than middle players. In the rest of the anthropometric measures and morphological components, there are no statistically significant differences between the players who play in different positions on the team [12]. Therefore, it seems that the morphological characteristics of top-level football players seem to be of great interest to some authors with the aim of finding the best morphological somatotype for particular levels of competition and player positions, although the specification depends on the development of technique and tactics [13]. It is important to consider that excess adipose tissue has a negative influence on sports performance, since the increase in body weight derived from fat is not accompanied by an increase in the capacity to produce greater force. Considering that acceleration is directly proportional to force but inversely proportional to body mass, excess adipose tissue at a given level of applied force will result in slower changes in speed [14].

On the other hand, the physiological demands of soccer require players to be trained in various components of physical conditioning, including aerobic capacity, speed, strength, power, and agility. Therefore, the assessment of physical condition in all its factors has been identified as an important contributor to selection in team sport athletes [15]. The implementation of appropriate, valid, and reliable aptitude tests is important to describe the profiles that fit the demands of the game [16]. In this sense, aerobic fitness tests such as the yo-yo Level 1 intermittent running test (Yo-Yo IR1) make it possible to evaluate the athlete's ability to recover from intense activity [17]. At the elite levels, female soccer players are reported to average 1500 m on the Yo-Yo IR1 [18]. Regarding the running speed in times of 10 m, the players of the Norwegian team ($2.17 \pm 0.06$ s) differed from the players of the first division ($2.21 \pm 0.07$ s) and the youth academies ($2.20 \pm 0.09$ s) and in their times for 40 m also (($6.12 \pm 0.02$ s) versus ($6.28 \pm 0.24$ s) and ($6.28 \pm 0.29$ s), respectively) [19]. Regarding the analysis by playing position, the forwards showed better times in 10 m ($2.16 \pm 0.07$ s) compared to the defenders ($2.19 \pm 0.06$ s), midfielders ($2.19 \pm 0.07$ s), and goalkeepers ($2.22 \pm 0.06$ s). However, Booysen et al. [20] reported that certain physical qualities do not differ according to playing positions, a finding that is observed in footballers in Europe and other countries. Consequently, the aerobic and anaerobic energy systems are used well during games, and training should increase the ability of players to perform high-intensity exercise and improve their ability to recover between these activities. While high-speed movement only contributes to about 11% of the total distance covered, high-intensity actions are likely to be performed at the most important moments of the game, for example, competing for ball possession or helping to score or avoid a goal. It seems logical, therefore, that players with better functional skills can perform more sprints, play longer at high

intensity, and have shorter recovery periods [21]. Research on performance in football has increased significantly in the last two decades [21,22]. Professionals and scientists related to football search for key optimization factors that contribute to improving the performance of elite and sub-elite players [2,4,21]. However, research on women's football is still significantly lower than that on men's football [21].

The aim of this study was to explore the anthropometric profile and physical fitness of members of the Chilean women's national football team, assess the presence of differences based on the playing position and analyze the possible correlations between the studied variables. Based on the cited studies, it was hypothesized that the goalkeepers are different from the rest of the footballers, in both anthropometric and physical fitness variables, whilst the field players (defenders, midfielders, or forwards) would exhibit differences in some physical fitness variables.

## 2. Materials and Methods

### 2.1. Design

This study followed a cross-sectional, descriptive, and correlational design.

### 2.2. Participants

Fifty players of the Chile women's national football team participated in this study, which was carried out during the period of training for the France 2019 Women's World Cup and the Japan 2020 Olympic Games—April 2019, first week. The age, height, and weight of the participants (mean ± SD) was 26.02 ± 4.04 years, 165.1 ± 8.1 cm, and 62.01 ± 4.5 kg, respectively, and they all had at least 7 years of experience in federal competitions. The participants were grouped according to their specific playing position: 7 goalkeepers (Gs), 14 defenders (Ds), 13 midfielders (Ms), and 16 forwards (Fs). They all had played in the senior category in different national leagues or competitions in different countries (Chile, Japan, Brazil, United States, Spain, and France). At the time when the tests were performed, the Chile women's national football team was 36th out of 155 according to the FIFA (International Federation of Association Football) women's world classification.

### 2.3. Temporalization

The selected participants carried out the physical condition tests in the first 4 days during a weekly training cycle (Figure 1). All players were required to avoid exhausting exercise 24 h prior to the tests, to prevent fatigue during the assessments. All the warm-up exercises were directed by the fitness coach. The participants performed a general standardized warm-up, which included slow running, multidirectional movements, and dynamic stretching, followed by a specific warm-up for each test, which lasted 15–20 min.

The different tests were scheduled as follows:

Day 1: anthropometric evaluation and squat test (ST).
Day 2: running speed, measured by the time to sprint 10 and 30 m (T10 and T30, respectively).
Day 3: countermovement jump (CMJ) and agility test (AGI).
Day 4: level-one yo-yo intermittent recovery test (MYYR1).

These tests are usually carried out by women's football teams [17,23–27], since they provide valid data to evaluate the anthropometry and physical condition of players. Therefore, all participants were familiar with the tests and performed them regularly. Each of the players was encouraged to apply their maximum effort on each test.

### 2.4. Procedure

#### 2.4.1. Anthropometry

The body mass index (BMI), which is an anthropometric indicator commonly used in studies with athlete populations [1,25,28,29], was calculated. The sum of six skinfolds (Σ6skinfolds: triceps, subscapular, supraspinal, abdominal, medial thigh, maximum calf), was used to estimate fat (%Fat) and muscle (%Muscle) percentage. These procedures have been validated by the International Society for the Advancement of Kinanthropometry

(ISAK). All measurements were recorded by an ISAK II–certified expert of the staff of the Chile Football Federation. The skin folds were measured using a slim-guide caliper (Rosscraft®, British Columbia, Canada) with 0.2 mm precision.

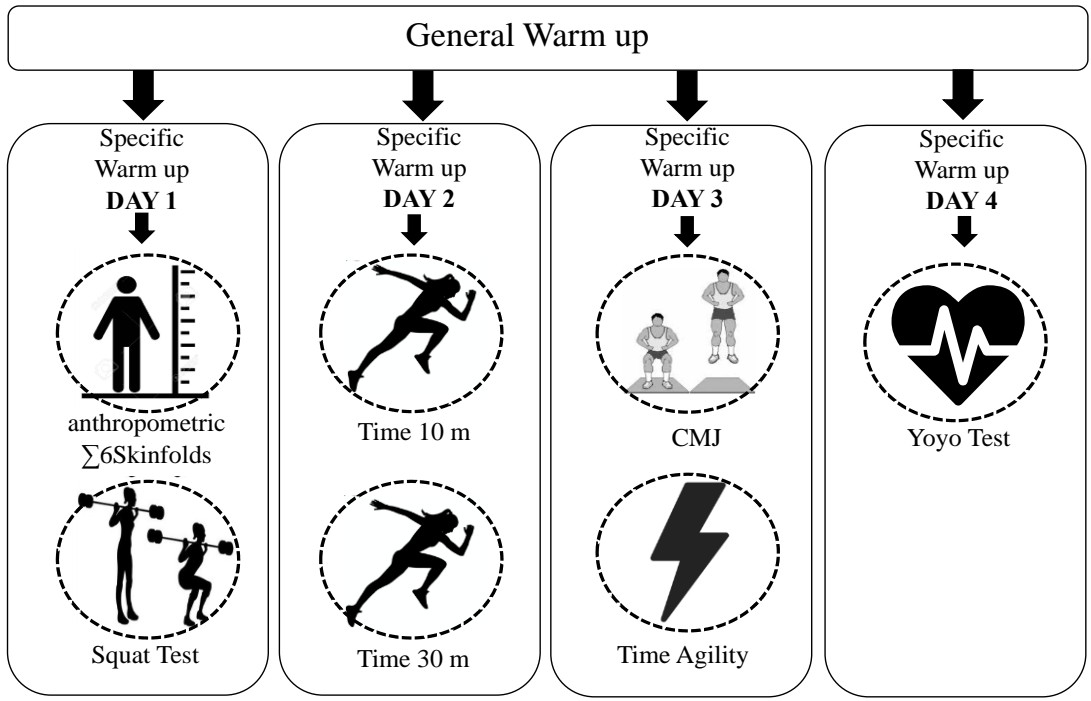

**Figure 1.** Distribution of anthropometric and physical condition evaluations.

### 2.4.2. Squat Test

This test has been validated to evaluate the levels of muscular strength in the lower limbs in football players [30–32]. Prior to the test, each participant performed a specific warm-up of three sets and three repetitions with a load of 20 kg. In the test, each participant performed five sets of squats (three repetitions per series), with progressively increasing loads of 20, 30, 40, 50, and 60 kg, and a 3 min rest between series. The concentric phase of the exercise was required to be executed at the maximum possible speed. None of the participants reached muscle failure in any of the sets. Three variables were recorded: (a) the average of the mean propulsive velocity for the five loads (MPV5l); (b) relative strength (RS), which was obtained from the quotient of one repetition maximum (1RM)/body weight; and (c) estimation of 1RM, which was determined from the mean propulsive velocity (MPV) of the last test load, which was in turn calculated from the formula proposed by [33]. The mentioned variables were determined using a linear encoder (Chronojump®, Barcelona, Spain).

The squat 1RM was calculated from the MPV with the last load using the following equation, Equation (1):

$$\%1RM = (-5.961\ VMP^2 - 50.71\ VMP + 117) \tag{1}$$

### 2.4.3. Time to Sprint 10 (T10) and 30 Meters (T30)

This test is commonly used to evaluate the acceleration capacity and maximum running speed of football players [34–36] (Figure 2). Before conducting the test, each player performed a specific warm-up of five progressive 30 m sprints. Three attempts of 30 m were carried out, with 3 min of rest between attempts. The test was conducted on a natural grass football pitch in the morning, at 15 °C and with a relative air humidity of 54%. The start position was standing, placing the leading foot right behind a line situated 0.5 m from the first photoelectric cell, in order to prevent the participant from blocking

the laser beam with her head or arms at the beginning of the run. Three photoelectric cells (Microgate®, Bolzano, Italy) were placed at the start, at 10 m, and at 30 m (Figure 2). The time of the three attempts was recorded in the following distances: 0–10 m (T10), as an indicator of acceleration, and 0–30 m (T30), as an indicator of maximum speed. The intraclass correlation coefficient (ICC) and the coefficient of variation (CV) were calculated, in order to establish the relative and absolute reliability, which were 0.88 and 5.05% for T10 and 0.98 and 4.48% for T30, respectively. The best T30 result was used in the analysis.

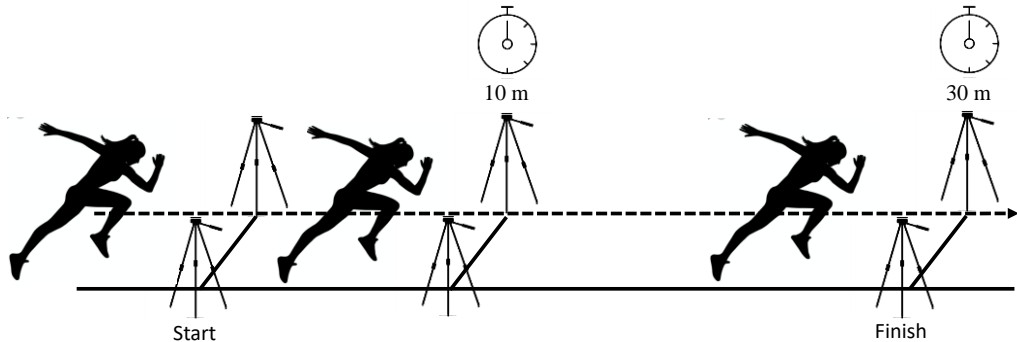

**Figure 2.** Acceleration evaluations at 10 m and velocity at 30 m.

### 2.4.4. Illinois Agility Test (IAT)

This test is frequently used to evaluate the agility and direction changes of football players [37,38]. Before conducting the test, each participant performed a specific warm-up of four submaximal-intensity attempts. For the test, the participants carried out three agility attempts on the natural grass pitch, with 3 min of rest between attempts. The shortest time of the three repetitions was selected. Photocells (Microgate®, Bolzano, Italy) were used to measure the time used in each attempt. The participants started in the supine position, with their feet situated 1 m behind the first beam (Figure 3). At the signal, they completed the tract as fast as they could. The ICC and CV were calculated to establish the relative and absolute reliability, which were 0.94 and 2.7%, respectively.

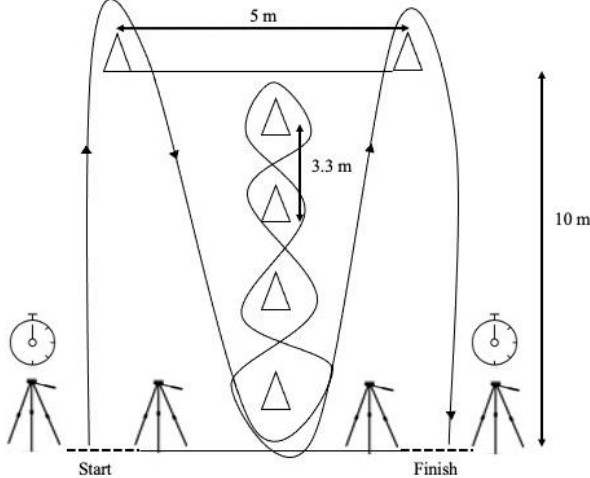

**Figure 3.** Illinois agility test.

### 2.4.5. Countermovement Jump (CMJ)

This test is usually conducted to measure the power of the lower limbs of female football players [27]. Before conducting the test, each player performed a specific warm-up of five jumps to a jump box, whose target surface was 40 cm from the ground. In the test, the participants carried out three attempts and for the CMJ, each participant started from

an upright standing position: the hands were placed on the hips throughout the test to eliminate any influence of arm swing. The participant rapidly squatted down until the knees were bent at approximately 90 degrees and then immediately jumped vertically as high as possible, landing on both feet at the same time [11]. This test was assessed by an Optojump Microgate® contact platform (Bolzano, Italy), with 3 min of rest between repetitions. The jump height (cm) was recorded, and the best result was selected. The ICC and CV were calculated to establish the relative and absolute reliability, which were 0.94 and 14.7%, respectively.

2.4.6. Level 1 Yo-Yo Intermittent Recovery Test (MYYR1)

This test is commonly employed to evaluate the intermittent resistance capacity of football players [17,39–42]. Before conducting the test, each player performed a specific warm-up of 10 progressive 20 m runs with direction changes. The participants conducted one attempt of the test, following its protocol [43]. For the statistical analysis, the meters covered (MYYR1) were recorded, and the maximal oxygen consumption ($VO_{2max}$), was calculated using the Equation (2) [43]:

$$VO_{2max} \; (mL/min/Kg) = (distance \; covered \; in \; MYYR1 \; (meters) \times 0.0084 + 36.4) \qquad (2)$$

*2.5. Ethical Considerations*

This type of intervention does not alter the normal football training or imply motor actions different from those of the usual practice of training sessions and matches. Moreover, all participants were subjected to a medical examination prior to the beginning of the season and carried out the tests with no injuries or physical discomfort. This study meets the requirements of the Declaration of Helsinki (WMA, 2013).

*2.6. Statistical Analysis*

For the descriptive analysis, the mean and standard deviation were calculated. The ICC and CV were determined to analyze the relative and absolute reliability of the variables that were measured more than twice. The Shapiro–Wilk test was performed to analyze whether the variables followed a normal distribution. Furthermore, the homogeneity of the variances was verified through the Levene test. To analyze the differences by playing position, a one-way ANOVA was applied. For the post hoc analyses between playing positions, the Tukey or Games–Howell test was used, depending on the homogeneity of the variances. The effect size (ES) was calculated using partial $Eta^2$ (0.01: small, 0.06: medium, 0.14: large). Correlations were calculated through Pearson's or Spearman's *r* depending on the normality of the variables, in order to study the degree of association between anthropometric and physical fitness variables. The *r* values were interpreted as trivial (0.00–0.09), small (0.10–0.29), moderate (0.30–0.49), large (0.50–0.69), very large (0.70–0.89), almost perfect (0.90–0.99), and perfect (1.0) [44]. The significance level was established at $p \leq 0.05$, and the confidence interval (CI) at 95% was calculated for all the measurements. Simple linear regressions were also calculated. The statistical analyses were conducted using SPSS IBM® software v.22 (New York, NY, USA).

**3. Results**

Table 1 shows differences between groups in weight ($p = 0.0001$) and height (0.002), with Gs presenting the greatest weight (66.7 ± 5.05 kg) and height (172.5 ± 6.6 cm) with respect to the rest of the players. There were also significant differences in %Fat ($p = 0.01$), with Gs also showing the greatest values, although, in this case, the differences were statistically significant only compared to Fs (28.12% vs. 22.87%). The variables age, %Muscle, and Σ6skinfolds did not show significant differences.

Table 2 shows the comparisons between groups for the physical fitness variables. As can be observed, there were differences in RS ($p = 0.001$), T10, T30, and AGI ($p < 0.001$), with Fs showing the best performance in these variables compared to the rest of the positions, whereas Gs showed the worst results. Differences were also identified in MYYR1 and

VO$_{2max}$ ($p < 0.001$), with Ms (1486.15 $\pm$ 235.42 m and 48.88 $\pm$ 1.97 mL/kg/min) presenting the best aerobic performance compared to the rest of the players. The variables MPV5l, 1RM and CMJ did not show statistically significant differences based on playing positions ($p > 0.05$).

**Table 1.** Anthropometric variables by playing positions. Mean $\pm$ standard deviation (M $\pm$ SD) and 95% confidence interval (CI 95%).

| Variables | Goalkeepers (n = 7) | | Defenders (n = 14) | | Midfielders (n = 13) | | Forwards (n = 16) | | Intergroup Comparisons | |
| --- | --- | --- | --- | --- | --- | --- | --- | --- | --- | --- |
| | M $\pm$ SD | CI (95%) | M $\pm$ SD | CI (95%) | M $\pm$ SD | CI (95%) | M $\pm$ SD | CI (95%) | *p* Value * | Effect Size ¥ |
| Age (years) | 25 $\pm$ 5.5 | (19.8–30.1) | 24.6 $\pm$ 4.6 | (21.9–27.3) | 24.9 $\pm$ 4.2 | (22.3–27.4) | 23.8 $\pm$ 4.5 | (21.4–26.3) | 0.921 | 0.010 |
| Weight (kg) | 66.7 $\pm$ 5.05 [abc] | (62–71.4) | 58.4 $\pm$ 4.4 [a] | (55.9–61) | 59.7 $\pm$ 4.8 [b] | (56.7–62.6) | 56.2 $\pm$ 5.3 [c] | (53.4–59.1) | <0.001 | 0.330 |
| Height (cm) | 172.5 $\pm$ 6.6 [abc] | (166.3–178.7) | 159.4 $\pm$ 4.2 [a] | (157–161.8) | 161.7 $\pm$ 4.5 [b] | (159–164.5) | 159 $\pm$ 5.8 [c] | (159–155.8) | 0.002 | 0.445 |
| %Fat | 28.12 $\pm$ 1.62 [a] | (26.61–29.6) | 26.49 $\pm$ 2.77 | (24.88–28.09) | 25.44 $\pm$ 2.77 | (23.76–27.1) | 24.24 $\pm$ 2.55 [a] | (22.87–25.6) | 0.010 | 0.216 |
| %Muscle | 45.93 $\pm$ 2.40 | (43.71–48.1) | 46.63 $\pm$ 2.72 | (45.05–48.2) | 47.16 $\pm$ 2.73 | (45.51–48.8) | 48.60 $\pm$ 1.99 | (47.54–49.6) | 0.067 | 0.143 |
| Σ6Skinfold | 67.78 $\pm$ 13.3 | (55.45–80.1) | 68.5 $\pm$ 14.3 | (60.19–76.8) | 64.15 $\pm$ 13.31 | (56.10–72.1) | 55.31 $\pm$ 13.92 | (47.89–62.7) | 0.057 | 0.150 |

\* ANOVA one-way. Post hoc pairwise Tukey or Games–Howell comparisons according to Levene test (two same letters in superscript indicate significant differences). ¥ Effect size by partial Eta$^2$ (0.01: small, 0.06: medium, 0.14: large).

**Table 2.** Physical fitness variables by playing positions. Mean $\pm$ standard deviation (M $\pm$ SD) and 95% confidence interval (CI 95%).

| | Goalkeepers (n = 7) | | Defenders (n = 14) | | Midfielders (n = 13) | | Forwards (n = 16) | | Intergroup Comparisons | |
| --- | --- | --- | --- | --- | --- | --- | --- | --- | --- | --- |
| | M $\pm$ SD | CI (95%) | M $\pm$ SD | CI (95%) | M $\pm$ SD | CI (95%) | M $\pm$ SD | CI (95%) | *p* Value * | Effect Size ¥ |
| MPV (m/s) | 0.97 $\pm$ 0.11 | (0.86–1.08) | 0.95 $\pm$ 0.09 | (0.87–1.01) | 0.96 $\pm$ 0.09 | (0.90–1.02) | 1.00 $\pm$ 1.11 | (0.94–1.06) | 0.608 | 0.039 |
| 1RM (kg) | 79.70 $\pm$ 8.29 | (72.0–87.3) | 82.08 $\pm$ 6.61 | (78.2–85.9) | 79.29 $\pm$ 6.56 | (75.32–83.25) | 82.35 $\pm$ 9.11 | (77.49–87.21) | 0.664 | 0.033 |
| RS (RM/BW) | 1.19 $\pm$ 0.13 [ab] | (1.07–1.31) | 1.41 $\pm$ 0.17 [a] | (1.31–1.51) | 1.33 $\pm$ 0.12 | (1.25–1.40) | 1.46 $\pm$ 0.13 [b] | (1.39–1.54) | 0.001 | 0.290 |
| T10 (s) | 2.06 $\pm$ 0.04 [abc] | (2.02–2.10) | 1.95 $\pm$ 0.09 [ad] | (1.89–2.00) | 1.95 $\pm$ 0.06 [be] | (1.91–1.99) | 1.86 $\pm$ 0.08 [cde] | (1.82–1.91) | <0.001 | 0.422 |
| T30 (s) | 5.12 $\pm$ 0.05 [abc] | (5.07–5.17) | 4.76 $\pm$ 0.12 [ad] | (4.69–4.83) | 4.84 $\pm$ 0.15 [be] | (4.75–4.93) | 4.61 $\pm$ 0.17 [cde] | (4.52–4.70) | <0.001 | 0.580 |
| IAT (s) | 17.7 $\pm$ 0.12 [abc] | (17.6–17.9) | 16.92 $\pm$ 0.33 [a] | (16.73–17.11) | 17.2 $\pm$ 0.33 [bd] | (17.00–17.41) | 16.81 $\pm$ 0.39 [cd] | (16.6–17.02) | <0.001 | 0.500 |
| CMJ (cm) | 30.6 $\pm$ 2.5 | (28.3–33) | 28.70 $\pm$ 4.41 | (26.15–31.25) | 28.08 $\pm$ 3.00 | (26.26–29.90) | 31.88 $\pm$ 5.24 | (29.08–34.67) | 0.076 | 0.137 |
| MYYR1 (m) | 902 $\pm$ 198.4 [abc] | (719.3–1086.4) | 1314.2 $\pm$ 238.8 [a] | (1176.3–1452.2) | 1486.1 $\pm$ 235.4 [b] | (1343.8–1628.4) | 1402.5 $\pm$ 297.3 [c] | (1244.05–1560.94) | <0.001 | 0.362 |
| VO$_{2 max}$ (mL/kg/m) | 43.9 $\pm$ 1.66 [abc] | (42.4–45.5) | 47.44 $\pm$ 2 [a] | (46.28–48.59) | 48.88 $\pm$ 1.97 [b] | (47.68–50.07) | 48.18 $\pm$ 2.49 [c] | (46.85–49.51) | <0.001 | 0.362 |

MPV: mean propulsive velocity; 1RM: one repetition maximum squat; RS: relative strength; T10: time to sprint 10 m; T30: time to sprint 30 m; CMJ: countermovement jump; IAT; Illinois agility test; MYYR1: meters in yo-yo test. \* ANOVA one-way. Post hoc pairwise Tukey or Games–Howell comparisons according to Levene test (two same letters in superscript indicate significant differences). ¥ Effect size by partial Eta$^2$ (0.01: small, 0.06: medium, 0.14: large).

Table 3 shows the correlations between anthropometric and physical fitness variables.

**Table 3.** Correlations between anthropometric and physical fitness variables.

| Variables | Weight | Height | %G | %M | ∑6fold | MPV | 1RM | RS | T10 | T30 | IAT | CMJ |
|---|---|---|---|---|---|---|---|---|---|---|---|---|
| **Height** | −0.178 | | | | | | | | | | | |
| **%Fat** | 0.418 ** | 0.302 * | | | | | | | | | | |
| **%Muscle** | −0.192 | −0.164 | −0.858 ** | | | | | | | | | |
| **Σ6Skinfold** | 0.527 ** | 0.112 | 0.823 ** | −0.627 ** | | | | | | | | |
| **MPV** | 0.182 | 0.075 | −0.248 | 0.210 | −0.202 | | | | | | | |
| **1RM** | 0.208 | −0.066 | −0.190 | 0.184 | −0.034 | 0.812 ** | | | | | | |
| **RS** | −0.648 ** | −0.563 ** | −0.498 ** | 0.313 * | −0.472 ** | 0.468 ** | 0.602 ** | | | | | |
| **T10** | 0.363 ** | 0.216 | 0.588 ** | −0.492 ** | 0.499 ** | −0.329* | −0.306 * | −0.507 ** | | | | |
| **T30** | 0.498 ** | 0.211 | 0.498 ** | −0.341 * | 0.466 ** | −0.199 | −0.195 | −0.541 ** | 0.777 ** | | | |
| **IAT** | 0.576 ** | 0.294 * | 0.421 ** | −0.249 | 0.422 ** | −0.279* | −0.303 * | −0.691 ** | 0.607 ** | 0.806 ** | | |
| **CMJ** | 0.058 | 0.167 | −0.394 ** | 0.377 ** | −0.385 ** | 0.530 ** | 0.487 ** | 0.336 * | −0.605 ** | −0.415 ** | −0.403 ** | |
| **MYYR1** | −0.470 ** | −0.195 | −0.343 * | 0.144 | −0.382 ** | −0.052 | −0.181 | 0.228 | −0.402 ** | −0.421 ** | −0.347 * | −0.067 |

** Correlation is significant at the 0.01 level (bilateral). * Correlation is significant at the 0.05 level (bilateral). MPV: mean propulsive velocity; 1RM: one repetition maximum squat; RS: relative strength; T10: time to sprint 10 m; T30: time to sprint 30 m; CMJ: countermovement jump; IAT: Illinois agility test; MYYR1: meters in yo-yo test.

Figure 4 shows the simple linear regressions between anthropometric variables and %Muscle. There were significant associations with %Fat and Σ6Skinfolds. Muscle percentage was associated with neither weight nor height.

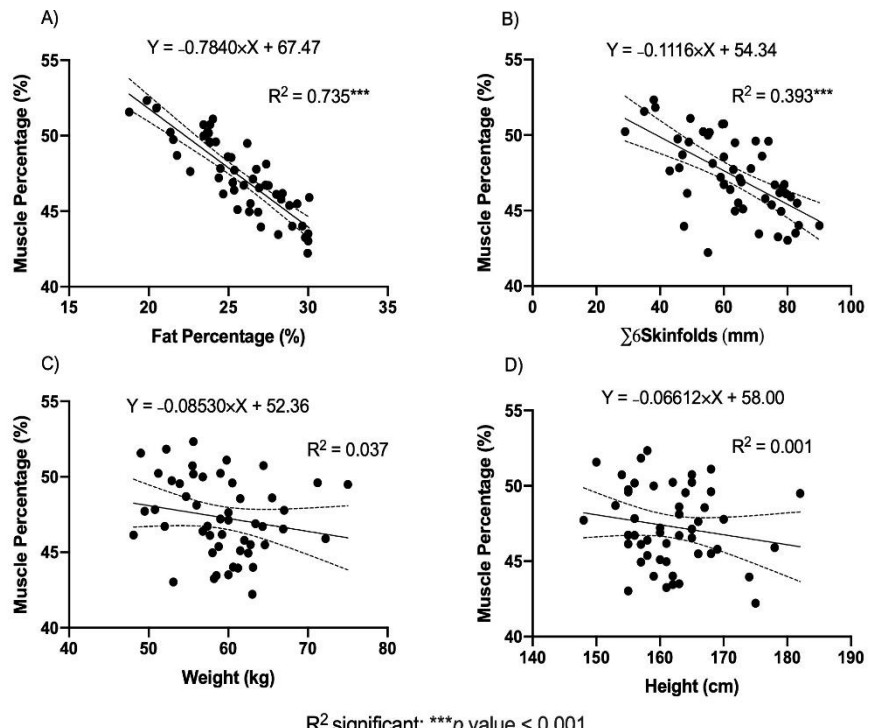

**Figure 4.** Simple linear regressions between muscle percentage and anthropometric variables (**A**): Fat Percentage; (**B**): Σ6Skinfolds; (**C**): Weight and (**D**): Height.

Figure 5 shows the simple linear regressions between RS and the variables running speed, IAT, and CMJ. Significant negative associations were identified with T10 ($R^2 = 0.242$), T30 ($R^2 = 0.293$), and IAT ($R^2 = 0.478$). On the other hand, there was a positive association with CMJ ($R^2 = 0.113$).

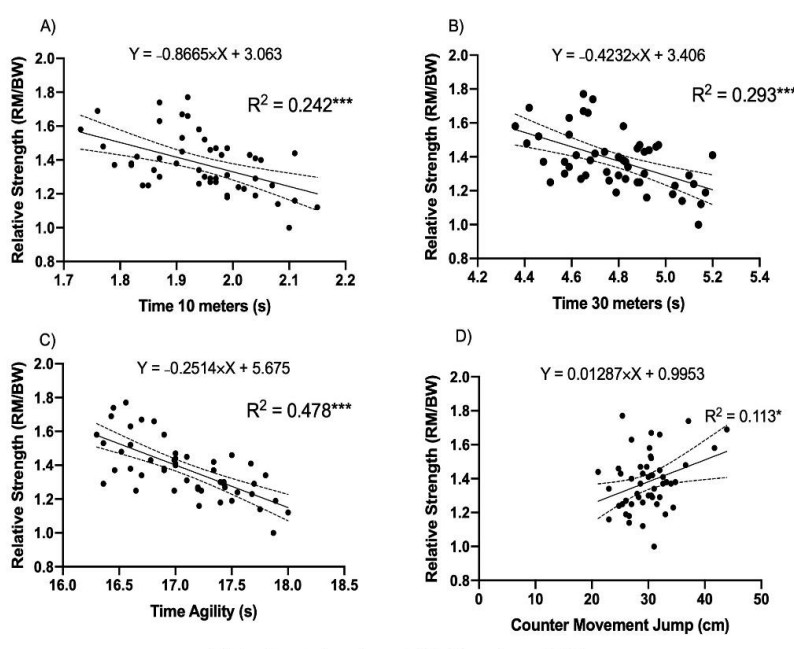

**Figure 5.** Simple linear regressions between relative strength and running speed (**A**,**B**), Agility (**C**), and CMJ (**D**).

Figure 6 shows the simple linear regressions between the anthropometric variables and RS. There were significant negative relationships with %Fat ($R^2 = 0.248$), Σ6Skinfolds ($R^2 = 0.222$), and height ($R^2 = -0.296$) and a positive relationship with %Muscle ($R^2 = 0.098$).

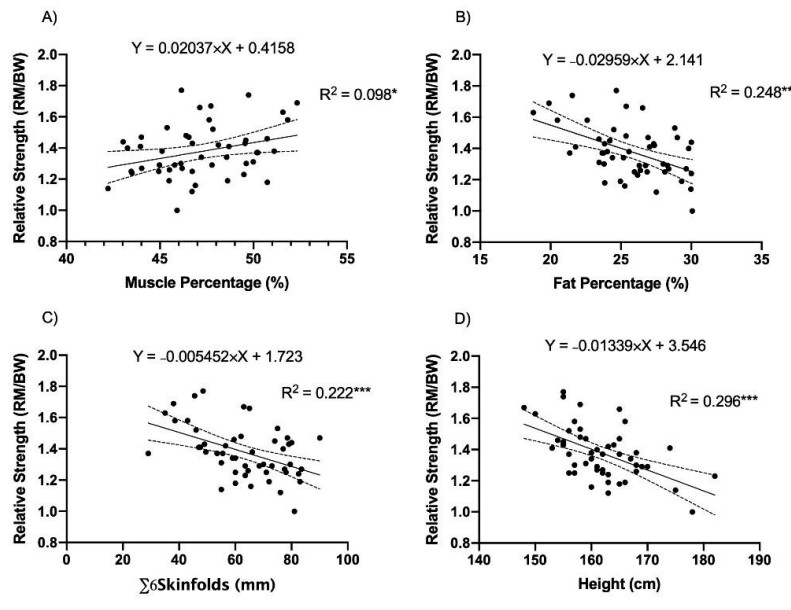

**Figure 6.** Simple linear regressions between relative strength and anthropometric variables (**A**): Muscle Percent; (**B**): Fat Percentage; (**C**): Σ6Skinfolds and (**D**): Height.

Figure 7 shows the simple linear regressions between the anthropometric variables and MYYR1. There were significant negative associations with %Fat ($R^2$ = 0.117–0.343), Σ6Skinfolds ($R^2$ = 0.146), and weight ($R^2$ = 0.221), which indicates that worse values of body composition are associated with lower aerobic performance. No statistically significant relationships were detected between MYYR1 and %Muscle.

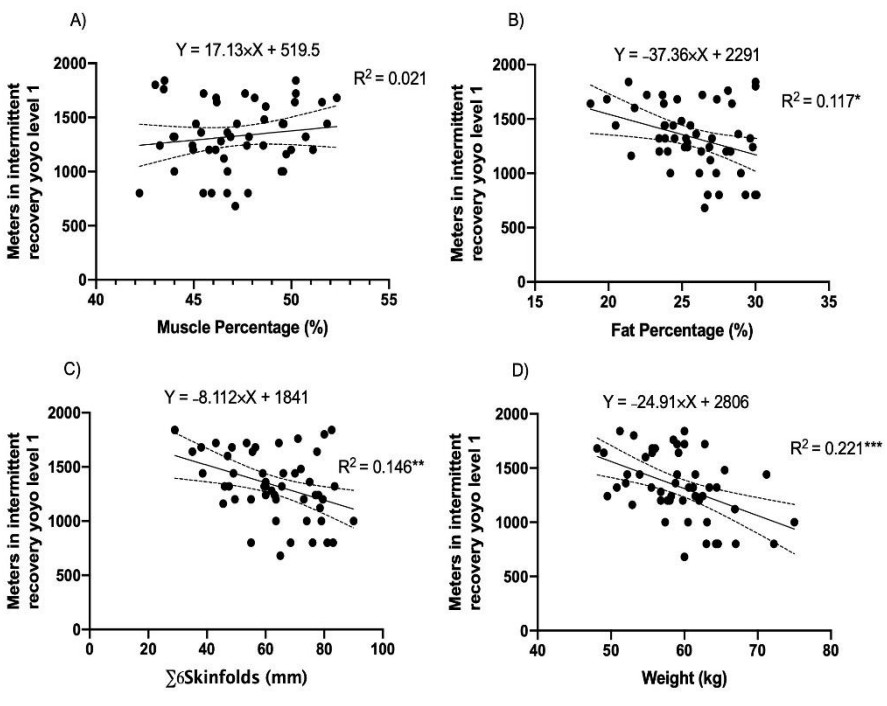

**Figure 7.** Simple linear regressions between yo-yo test and anthropometric variables (**A**): Muscle Percent; (**B**): Fat Percentage; (**C**): Σ6Skinfolds and (**D**): Weight.

## 4. Discussion

The aim of this study was to describe, compare, and associate the anthropometric profile with the level of physical fitness in the members of the Chile women's national football team, differentiating the participants by their playing position.

The findings of this study in the anthropometric variables (Table 1) indicate that there were differences between playing positions in weight ($p < 0.001$) (ES = 0.330) and height ($p = 0.002$) (ES = 0.445), with Gs showing the greatest values in weight and height compared to the rest of the players. These data are in line with those reported in the report on the Canada 2015 World Cup [45], which describes that Gs (vs. field players) were the heaviest (67 ± 5.2 vs. 59.8 ± 6.1 kg) and tallest (173.6 ± 4.1 vs. 166.4 ± 6.6 cm). Something similar was found by Kammoun et al. [11] with first-division players of the Tunisian league, where the goalkeepers presented a significantly higher difference ($p < 0.001$) in weight and height. However, these results are not in agreement with those obtained by Lockie et al. [46], who found no differences between football players of different playing positions in weight and height ($p > 0.05$). This could be due to the fact that their participants were division I collegiate female soccer players and our participants were elite players. Moreover, the mentioned study only analyzed 3 Gs, which was probably insufficient to detect statistically significant differences. Therefore, these results seem to indicate that the differences in body composition based on the playing position could respond to the differences in the specific demands between goalkeepers and field players in maximum sport conditions, such as in elite competitions [45]. Similar results were found with respect to %Fat, showing differences between playing positions ($p = 0.010$) (ES = 0.216) (Table 1), with Gs obtaining the greatest levels of %Fat; in this case, significant differences were detected between

Gs and Fs. In a recent study with professional football players, significant differences were reported ($p = 0.036$) in %Fat between players, with Gs showing the greatest %Fat values compared to the rest of the players [3], the same findings presented by Kammoun et al. [11] where the outfield players had significantly ($p < 0.001$) lower fat percentages than the goalkeepers. Despite this study, as well as our findings, no significant differences were observed in body fat between field players. Regarding %Muscle and Σ6skinfolds, no differences were found based on playing position ($p = 0.067$ and $p = 0.057$, respectively). Similar results were obtained by Nikolaidis [3] and Barraza and colleagues [14], who found no differences ($p > 0.05$) in %Muscle or Σ6skinfolds in football players. These results could be due to the fact that high-performance football players require certain muscle mass levels to respond to high levels of training stimuli and competitive demand [4]. To sum up, the most significant finding recorded in this study regarding the anthropometric variables is that there were no differences between the field players (Ms, Ds, and Fa); that is, only Gs exhibited differences with respect to the rest of the players.

Table 2 shows the results obtained in the physical fitness tests, compared by playing position. Regarding RS, although Fs obtained the highest levels in this variable, there were no differences with the field players (Ds and Ms). However, Gs showed significantly lower values compared to Ds and Fs ($p = 0.001$) (ES = 0.290). RS is a variable that considers the generation of strength as a function of body weight (RM/BW) [47]. This is why Gs obtained the lowest RS, since they usually present the greatest values in body weight [45].

Regarding performance in running speed, differences were detected between playing positions ($p < 0.001$) in T10 (ES = 0.422), T30 (ES = 0.580), and AGI (ES = 0.500). Gs obtained significantly worse results than the rest of the playing positions. Similarly, Fs obtained the best performance in all the displacement variables, suggesting that they are the fastest players. These findings are similar to those found in selected players from Norway [19]. Regarding the analysis by playing position, the forwards showed better times in 10 m (2.16 ± 0.07 s) compared to the defenders (2.19 ± 0.06 s), midfielders (2.19 ± 0.07 s), and goalkeepers (2.22 ± 0.06 s). These results are not in agreement with those reported by Lockie et al. [46], who found no differences between playing positions in T10, T30, or AGI. These differences between studies could be due to the fact that the mentioned study analyzed a sample of division I collegiate female soccer players, thus there would not be excessive specialization in the performance demands by playing positions at that competition level [15]. That is, the findings presented in this study could be explained by the fact that, at a high competition level, there are differences in the specific demands by playing positions, such as scoring goals (Fs) and preventing goals (Ds), which usually require high-intensity displacement actions at maximum speed, in order to anticipate to the opponents in very fast actions that take place in short distances [48].

CMJ showed no statistically significant differences between playing positions ($p = 0.076$) (Table 2). The scientific literature shows contradicting findings. The study of Booysen et al. [20] on selected South African elite players did not find significant differences between playing positions for CMJ height ($p > 0.05$). Something similar was also reported by Romero-Caballero and colleagues [49], who did not detect differences in the CMJ test in female football players ($p = 0.848$). However, Sedano et al. [50] did show differences between positions ($p < 0.05$), probably due to the fact that their participants were non-elite players, something similar was seen in the study by Kammoun et al. [11] with Tunisian players, showing differences in CMJ between outfield players ($p < 0.05$). Therefore, it is possible that the value of this variable does not seem to distinguish between playing positions in elite football players. Lastly, the study of Datson et al. [22] suggests that a CMJ value above 34.4 cm could be considered as superior jumping capacity, and a CMJ value above 29.8 cm could be considered as a threshold measure to discriminate between competitive levels among elite players.

Regarding MYYR1 (which allows estimating $VO_{2max}$), there were differences between positions ($p < 0.001$) (ES = 0.362) (Table 2). In this test, Gs showed a significantly lower performance than the rest of the players. Ms obtained the best values, although these

were not statistically better than those of the other field players (Fs and Ds). These results are similar to those of Kammoun et al. [11]. In this study, the goalkeepers covered less distance in the MYYR1 than other position players ($p < 0.001$). However, no significant differences were found between the other field positions. These results are not in line with those reported by Lockie et al. [46] and Risso et al. [15], who found no differences between playing positions ($p > 0.05$) in this variable. These differences could be since their participants were collegiate players, as well as due to the small number of Gs (3). Similarly, the level of specialization in the competitive requirements may not be as high in the collegiate category [15]. However, in the elite category, physical skills and aptitudes must respond to a specific pattern of demands in competition [8]. The differences observed in our study could be due to the fact that Ms and Fs are required to be faster, in order to elude the opponent and score goals [51], and that Ms are precisely the ones who cover the longest distances in high-competition matches [5]. For example, according to the contribution of Bangsbo et al. [43], the authors categorize the performance of elite soccer players by MYYR1 performance level in top elite (1600 m), moderate elite (1360 m), and sub-elite (1160m), respectively. Therefore, according to our results, the Gs (902.85 $\pm$ 198.47 m) would be classified as sub-elite, Ds (1314.28 $\pm$ 238.89 m) as moderate elite, and finally, Ms (1486.15 $\pm$ 235.42 m) and Fs (1402.50 $\pm$ 297.35 m) as superior elite. In the same sense, our records are slightly lower than those reported by Doyle et al. [18] with selected adult female players from Ireland with an average of 1500 m on the MYYR1.

Figures 4–7 show the linear regression between the studied variables, where significant relationships can be observed between the anthropometric and physical fitness variables (Table 3).

Figure 4 presents the relationships between %Muscle and the anthropometric variables. As can be observed, %Muscle significantly explains the changes in %Fat ($R^2 = 0.735$) and $\Sigma$6skinfolds ($R^2 = 0.393$). This parameter of body composition, which indicates muscle mass, has been proved to be a determining variable in the anthropometry of Chilean [14,24] and international players [4]. In this sense, correlations have been found between field tests (e.g., jump height and shooting speed with dominant and non-dominant leg) and muscle mass, height, and leg length [50].

Figure 5 shows the relationships between RS and performance in running speed, agility, and jump height (T10 $R^2 = 0.242$, T30 $R^2 = 0.293$; IAT $R^2 = 0.478$; CMJ $R^2 = 0.113$), with all showing statistical significance ($p < 0.05$). Thus, it is demonstrated that RS is significantly associated with the results of tests that measure actions of explosive strength and maximum intensity, i.e., the greater the performance in explosive tests, the better the results in RS. These findings are similar to those reported by Marcote-Pequeño et al. [52], who evaluated the force–velocity profile of professional female football players of the Spanish league. These authors obtained strong correlations (r = 0.751) in maximum speed between the jump test (squat jump) and the running test (T20). Similar results have been reported by Jiménez-Reyes et al. [53], who, with the same protocol, recorded strong relationships in amateur players (r = 0.622) and professional players (r = 0.492). These results suggest that the jump and running speed or agility tests could provide similar information, particularly with respect to the generation of maximum strength (in this case, relative maximum strength). These tests are similar to the most frequent actions in certain situations in a football match [48]. Likewise, the data of this study are in line with those reported by Emmonds and colleagues [54], who concluded that different manifestations of strength show moderate and strong relationships with performance in displacement tests.

Figure 6 presents how the performance in RS can explain the changes in the anthropometric variables, all $R^2$ values are statistically significant ($p < 0.05$). These results suggest that body composition could explain some of the physical fitness variables in the studied players. Thus, for example, it has been reported that female football players with lower %Fat develop high rates of strength generation [3], which is in line with the results obtained in the present study. These findings are also in agreement with those shown by Borcherie et al. [55] in international athletes, where the players who obtained lower %Fat and greater

%Muscle obtained better correlations in the actions that involved a greater generation of strength to jump and run.

Figure 7 shows the relationships between MYYR1 and anthropometric variables. All $R^2$ values are statistically significant ($p < 0.05$), except for %Muscle ($R^2 = 0.021$). These results are similar to those presented by Mujika et al. [56], who reported that the football players who obtained greater performance in MYYR1 showed lower body fat percentage [56]. These findings are also similar to those presented by Bajramovic et al. [57], who showed that, among the Bosnian players who competed at the international level, a greater oxygen consumption was correlated with a lower body weight.

## 5. Conclusions

One of the most evident findings of this study is that Gs are different from the rest of the players, in both anthropometric and physical fitness variables. Gs present greater height and weight, although there are no differences in %Muscle between playing positions. On the other hand, the three groups of field players (Ds, Ms, and Fs) showed no significant differences between them in any of the anthropometric variables studied. Similarly, Gs showed significantly worse performance results than the rest of the players in running speed (T10 and T30), agility (IAT), and aerobic power (MYYR1). The most remarkable finding, when comparing the results obtained in the physical fitness tests by playing positions, is that Fs proved to be the fastest players, showing better performance in the running speed tests than the other three groups.

In the Chile women's national football team, it was found that better results in body composition variables (high %Muscle and low %Fat) are associated with better results in indicators of physical fitness, such as relative strength, whereas %Fat, weight, and Σ6skinfolds have a negative influence on performance in estimated $VO_{2max}$. The relative strength results of the studied players showed a strong correlation with their performance in the explosive actions tests (T10, T30, and IAT). This research analyzes only a single team, which is a clear limitation, so more studies with larger samples would be necessary to corroborate these results.

## 6. Practical Applications

The anthropometric characteristics and physical fitness level of female players have been shown to be important attributes for soccer performance. This study provides baseline data on elite female players of different playing positions that can be used to guide the development of systems, monitoring, selection, and evaluation of the effectiveness of training programs. Our data suggest that female players should aim to implement individualized training for the development of optimal anthropometric and fitness values to compete at the elite level.

**Author Contributions:** Conceptualization, R.V.-V. and J.A.G.-J.; data curation, S.Z. and E.M.-S.; formal analysis, E.M.-S. and J.A.G.-J.; investigation, S.Z. and R.V.-V.; methodology, R.V.-V. and J.A.G.-J.; resources, R.V.-V. and E.M.-S.; supervision, E.M.-S. and J.A.G.-J.; visualization, S.Z. and R.V.-V.; writing—original draft, E.M.-S., S.Z., R.V.-V., and J.A.G.-J.; writing, reviewing, and editing, R.V.-V. and J.A.G.-J. All authors have read and agreed to the published version of the manuscript.

**Funding:** This research received no external.

**Institutional Review Board Statement:** This intervention does not alter the normal football training or imply motor actions different from those of the usual practice of training sessions and matches. Moreover, all participants were subjected to a medical examination prior to the beginning of the season and carried out the tests with no injuries or physical discomfort. This study meets the requirements of the Declaration of Helsinki (WMA, 2013).

**Informed Consent Statement:** Informed consent was obtained from all subjects involved in the study.

**Data Availability Statement:** The data presented in this study are available on request from the corresponding author. The data are not publicly available due to privacy.

**Acknowledgments:** To players, coaching staff, and to the Federación de Fútbol de Chile for facilitating the development of this research.

**Conflicts of Interest:** The authors declare no conflict of interest.

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
