# Peer review of "Anthropometric Profile and Physical Fitness Performance Comparison by Game Position in the Chile Women’s Senior National Football Team"

_applsci, doi:10.3390/app11052004_

Round 1
Reviewer 1 Report
The paper is well written and is interesting.
The introduction can be improved in the background of the study.
Regarding material and methods are well reported and clear.
The results are well reported and clear.
The references are correct and don't are detected inappropriate self-citations by authors
In conclusion the paper is good with a minor revision regarding background of introduction.
Reviewer 2 Report
I have carefully reviewed manuscript of Villaseca‐Vicuña et al. titled: "Anthropometric Profile and Physical Condition Performance Comparison by Game Position in the Chile Womenʹs Senior National Football Team".
The authors have conducted research on 50 women football national team players from Chile. They have been subjected to standard anthropometric measures and physical condition tests to examine their conditioning status and differences in this status regarding to playing positions. Introduction needs to be addressed better to aim and variables of the study. Methods and results section is written in a good way. Discussion could be written more flowingly, but it also depends on authors ' style of writing so this is just a note/hint, not recommendation. Conclusion is well written and authors have reflected to all significant results of the study, but need to add some practical implications for practice of this study.
Introduction
The rationale for the study should be better explained for readers, also the aims and essence of You theme of this article should be referenced and shown better. In my opinion it is too short and You write more about football and known facts than Your aim of study.
Try to find some new references as the newest is 5 years old and there are many good quality papers regarding Your article and Your subjects.
With the aim of the study which is correctly written add what is/are Your hypothesis?
Methods
Please correct here and throughout of Your article decimal places describing subjects variables such as age, weight, ... (if they are really full numbers - ignore my line here).
Just a hint, "p" should be in italic in Methods section as in remainder of article.
Add equation in procedure subsection where You mention the sum of 6 skin‐folds. Explain how it is calculated and reference this.
Results
I have no complaints about Results section - author did a very good job.
Discussion
You are comparing Your results of professional players with the results of amateur players in several variables. Try to find articles with professional players as subjects and compare Your results with those.
Try to consider adding a line in Your discussion about the limitations and strengths of this study which could be interesting for readers.
Conclusions
Well written according to results and discussion of this article, but try to focus also on practical implications for practice of this study to make better effect of conclusions.
Reviewer 3 Report
Dear authors, congratulate you for carrying out the study and the sample used, since they are high-level players. However, different parts of the study have to be significantly modified so that it can be considered for publication.
Introduction section:
The purpose of this study was to explore the anthropometric profile and conditioning variables of
members of the Chile women’s national football team, assess the presence of differences based on the
playing position and analyze the possible correlations between the studied variables, however in discussion section the purpose its different.
An adequate contextualization of the competition demand for female soccer players is made, but that is not the purpose of the study (demands/demarcations/anthropometric/conditioning performance) and aspects such as anthropometric characteristics or physical performance and assessment tests are not contextualized, so the study purpose is not properly entered.
In line 50 the demands of the match are indicated according game position, but studies have shown that the physical demands and bioenergetics across positions vary markedly according to the precise tactical role of the player (wide vs. central positions). Have the authors considered this aspect?
Materials and Methods section:
Line 79: All the players took the tests of condition performance at the same time? At what point in the competitive season were the conditioning tests carried out?
Figure 1 duplicates information manifested in the text, determine if in text or in figure but not duplicate.
Line 121: Why did the authors select the 30 m distance to assess the maximum velocity of the players? Precious studies show that velocity is evaluated in 10 meters faster over 40 meters
Mendez-Villanueva A, Buchheit M, Simpson B, Peltola E, Bourdon P. Does on-field sprinting performance in young soccer players depend on how fast they can run or how fast they do run? J Strength Cond Res. 2011;25(9):2634–8.
Line 123: The references used to justify the maximal velocity test do not use the distance used in the study.
Line 139: Are there references that justify the use IAT in female soccer players? In addition, the references provided use the modified IAT. Have the authors taken this into account both in the application of the test and in the discussion section to compare the performance? The authors have to take into account that the time will be different
Line 154: Can the authors justify the large CV in CMJ test?
Line 153: Consider authors this manuscript: Martínez-Lagunas V, Hartmann U. Validity of the Yo-Yo Intermittent Recovery Test Level 1 for Direct Measurement or Indirect Estimation of Maximal Oxygen Uptake in Female Soccer Players. Int J Sports Physiol Perform. 2014;9(5):825–31.
Figure 1 and figure 2 can be suppressed indicating the dimensions in the text.
Results section:
Line 191: The variables age, %muscle and Σ6skinfolds did not show significant variables. What are the authors referring to “significant variables”?
In table 1 and table 2 the authors do not indicate the meaning of the letters used to indicate significant differences.
Why do the authors duplicate the description of correlation results in the Table 3 and in the subsequent figures?
Line 214: The authors have performed a correlation not a linear regression.
Discussion section
Several paragraphs of the discussion seem more like results than discussion (Line: 262, 300, 303, 309, 334) it is necessary to discuss the results and the modified wording so as not to duplicate with the results section.
Line 243: participant of Lockie (34) are amateurs or collegiate?
Line 250: Why do the authors consider that G obtaining the greatest levels of adipose tissue?
Line 268-271: In male soccer player F has worse physical performance than other positions, what can be the difference with female soccer players?
Bradley PS, Mohr M, Bendiksen M, Randers MB, Flindt M, Barnes C, et al. Sub-maximal and maximal Yo-Yo intermittent endurance test level 2: heart rate response, reproducibility and application to elite soccer. Eur J Appl Physiol. 2010/11/18. 2011;111(6):969–78.
Krustrup, P., Mohr, M., Amstrup, T., Rysgaard, T., Johansen, J., Steensberg, A. et al. (2003). The yo-yo intermittent recovery test: Physiological response, reliability, and validity. Medicine and Science in Sports and Exercise, 35, 697–705. Krustrup,
Krustrup, P., Mohr, M., Nybo, L., Jensen, J.M., Nielsen, J.J., & Bangsbo, J. (2006a). The Yo-Yo IR2 test: Physiological response, reliability and application to elite soccer. Medicine and Science in Sports and Exercise, 38, 1666–1673.
Line 280-284: CMJ and T10 and T30 depend on neuromuscular factors, why is the justification for them different depending on whether they are amateur or not? In other words, the authors indicate that T30 is not different in amateurs due to lack of specialization, while CMJ is different in Sedado et al., Because they are amateurs. Why?
Line 318-320: A relationship is not always causal, modify the writing of the paragraph.
Reviewer 4 Report
The aim of the current study was to identify associations between physical and performance characteristics of elite female soccer players. The study contains practical insight regarding associations that may be used to improve performance. My comments are listed below.
It may be helpful for the global audience to distinguish between football, American football, and soccer. Even if it's just once in the introduction, this may help the lay reader.
The height is reported in M but is listed as CM.
Do the authors have any reliability data from the skinfold assessments. This test is very technique specific and as many conclusions are drawn from this test, the overall validity of the research team's precision is of significance. If not, this should be addressed as a limitation.
Above comment goes for additional measures not reporting reliability data (e.g., squat test).
More details are needed for the CMJ protocol. For example, did the participants include an arm swing (Heishman et al.,) has documented the importance of this and this could potentially influence the outcomes of the current data.
Please include the specific ICC that was used.
Please include the calculation for effect size.
Throughout the results there are a few places where the authors have different spacing approaches for the included p-values. Please adjust to remain consistent throughout.
I think the word "differences" instead of variables should be used for the last word in the first paragraph of results?
In the tables, please keep the decimal places consistent.
For the correlations, could you also please include the coefficient of determination. However, these figures present the data quite nicely- great work.
You can't have a p-value of 0.0 please adjust throughout the manuscript.
The discussion needs to be written more concisely- there were a few places that propositions were made that were not sound (e.g., Line 256; Line 272; Line 285). Please reformat the discussion and discuss the most pertinent observations and refrain from restating the results section.
Round 2
Reviewer 2 Report
I am satisfied with Your updated manuscript - (almost*) all concerns and comments were solved and manuscript is now much better quality-wise.
* - please check following comment and correct in WHOLE manuscript:
Just a hint, "p" should be in italic in Methods section as in remainder of article.
Author Response
* - please check following comment and correct in WHOLE manuscript:
Just a hint, "p" should be in italic in Methods section as in remainder of article.
Thank you for your comment. Corrected.
Reviewer 3 Report
Dear authors, congratulate you for the improvements made in the manuscript, which undoubtedly improve the quality and understanding on for the readers.
Regarding my comment from the first review: " Line 79: All the players took the tests of condition performance at the same time? At what point in the competitive season were the conditioning tests carried out?"
It is important to indicate exactly at what time of the season the data were collected: end of season without regular domestic competition, in the different windows of international matches.... Different studies have analyzed fitness test performance throughout the season and it is essential to determine this aspect to contextualize the data. Your explanation in the manuscript is not enough.
The references included to justify the IAT are now correct, although the description uses the reference from Vescovi & McGuigan, 2008 which uses a different distance, I propose to delete it so as not to confuse readers.
Regarding my comment on linear regression: It is true that a figure improves readers' understanding, but it is not correct to duplicate results in tables and figures without a more solid justification of why it has been carried out. :
-The authors do not explain the procedure in the statistical analysis section (regresion).
-In figures, the authors report data of correlation (r) not regression (r2), therefore my comment of duplicating information and not reported p value of regression.
-Correlation quantifies how related two variables are, while linear regression consists of generating an model that, based on the relationship (correlation) between both variables, allows predicting the value of one from the other. Why a regression without significant correlation?
-From my point of view, in this study it would be more interesting to perform a multiple regression to try to determine which anthropometric variable predicts performance in the highest percentage
Line 342: correct .p=
Best regards
Author Response
Regarding my comment from the first review: " Line 79: All the players took the tests of condition performance at the same time? At what point in the competitive season were the conditioning tests carried out?" It is important to indicate exactly at what time of the season the data were collected: end of season without regular domestic competition, in the different windows of international matches.... Different studies have analyzed fitness test performance throughout the season and it is essential to determine this aspect to contextualize the data. Your explanation in the manuscript is not enough.
Thank you for your comment. Corrected.
The references included to justify the IAT are now correct, although the description uses the reference from Vescovi & McGuigan, 2008 which uses a different distance, I propose to delete it so as not to confuse readers.
Thank you for your comment. Deleted
Regarding my comment on linear regression: It is true that a figure improves readers' understanding, but it is not correct to duplicate results in tables and figures without a more solid justification of why it has been carried out. :
-The authors do not explain the procedure in the statistical analysis section (regresion).
Corrected
-In figures, the authors report data of correlation (r) not regression (r2), therefore my comment of duplicating information and not reported p value of regression.
Corrected
-Correlation quantifies how related two variables are, while linear regression consists of generating an model that, based on the relationship (correlation) between both variables, allows predicting the value of one from the other. Why a regression without significant correlation?
Corrected.
-From my point of view, in this study it would be more interesting to perform a multiple regression to try to determine which anthropometric variable predicts performance in the highest percentage
Thanks for your suggestion. In this research, there is a lot of data, so there are many possible multiple regressions. In this case, we think that the results recorded in simple linear regressions are easier to interpret and easier to apply in practice.
Line 342: correct .p=
Corrected.
Reviewer 4 Report
I would like to acknowledge the author's responses to the suggested revisions. Overall, they did a great job and I believe this work will provide scientific merit to the applied sports performance realm.
Congratulations on the great work and best of luck in your future endeavors.
Author Response
Thank you for your comment.